# A prospective survey of *Streptococcus pyogenes* infections in French Brittany from 2009 to 2017: Comprehensive dynamic of new emergent *emm* genotypes

Sarrah Boukthir[1,2,3☉], Séverine Moullec[2,3☉], Marie-Estelle Cariou[4], Alexandra Meygret[1,3¤], Jeff Morcet[1,2], Ahmad Faili[2,5], Samer Kayal[1,2,3] *

**1** CHU de Rennes, Service de Bactériologie-Hygiène Hospitalière, Rennes, France, **2** Inserm, CIC 1414, Rennes, France, **3** Université Rennes 1, Faculté de Médecine, Rennes, France, **4** Centre Hospitalier de "Bretagne Sud", Laboratoire de Biologie, Lorient, France, **5** Université Rennes 1, Faculté de Pharmacie, Rennes, France

☉ These authors contributed equally to this work.
¤ Current address: Centre Hospitalier de la Basse-terre, Service de Bactériologie, Basse-terre, Guadeloupe
* samer.kayal@univ-rennes1.fr

**Data Availability Statement:** All relevant data are within the manuscript and its Supporting information files.

## Abstract

*Streptococcus pyogenes* or group A *Streptococcus* (GAS) causes diseases ranging from uncomplicated pharyngitis to life-threatening infections. It has complex epidemiology driven by the diversity, the temporal and geographical fluctuations of the circulating strains. Despite the global burden of GAS diseases, there is currently no available vaccination strategy against GAS infections. This study, based on a longitudinal population survey, aimed to understand the dynamic of GAS *emm* types and to give leads to better recognition of underlying mechanisms for the emergence of successful clones. From 2009 to 2017, we conducted a systematic culture-based diagnosis of GAS infections in a French Brittany population with a prospective recovery of clinical data. The epidemiological analysis was performed using *emm* typing combined with the structural and functional cluster-typing system for all the recovered strains. Risk factors for the invasiveness, identified by univariate analysis, were computed in a multiple logistic regression analysis, and the only independent risk factor remaining in the model was the age (OR for the entire range [CI$_{95\%}$] = 6.35 [3.63, 11.10]; *p*<0.0001). Among the 61 different *emm* types identified, the most prevalent were *emm*28 (16%), *emm*89 (15%), *emm*1 (14%), and *emm*4 (8%), which accounted for more than 50% of circulating strains. During the study period, five genotypes identified as *emm*44, 66, 75, 83, 87 emerged successively and belonged to clusters D4, E2, E3, and E6 that were different from those gathering "Prevalent" *emm* types (clusters A-C3 to 5, E1 and E4). We previously reported significant genetic modifications for *emm*44, 66, 83 and 75 types resulting possibly from a short adaptive evolution. Herein we additionally observed that the emergence of a new genotype could occur in a susceptible population having specific risk factors or probably lacking a naturally-acquired cluster-specific immune cross-protection. Among emergent *emm* types, *emm*75 and *emm*87 tend to become prevalent with a stable annual incidence and the risk of a clonal expansion have to be considered.

**Funding:** This work was supported by the University Rennes-1 Medical School, and by University Hospital Center of Rennes-France and did not receive any specific grant from funding agencies in the public, commercial, or not-for-profit sectors.

**Competing interests:** The authors have declared that no competing interests exist.

## Introduction

*Streptococcus pyogenes* or Group A *Streptococcus* (GAS) are Gram-positive cocci that usually colonize the human skin and throat and cause a wide variety of diseases ranging in severity from uncomplicated pharyngitis to severe and life-threatening infections [1]. On a global scale, GAS ranked as the fourth deadliest bacterium in the world, with more than 500,000 deaths per year [2].

Lancefield's pioneering work demonstrated that GAS infections elicit a robust immune response by producing opsonizing antibodies against the cell surface M protein encoded by the *emm* gene [3]. For GAS, M protein is a major immunological and virulence determinant able to bind several host factors (fibrinogen, plasminogen, immunoglobulins) [1]. For an epidemiological survey of GAS infections, *emm* genotyping based on the sequence of the 5' hypervariable end of the *emm* gene, is a worldwide-accepted marker [4]. More than 250 different genotypes have been identified and referenced by the Centers for Disease Control and Prevention (CDC), Atlanta. The M-protein-based vaccine appears to be the most promising strategy. Although many trials are in progress (CANVAS Group: Coalition to Accelerate New Vaccines Against *Streptococcus*) [5], there is currently no available vaccination against GAS infections. Its unavailability is mainly explained by the epidemiological complexity of circulating strains stressed by their geographical diversity and temporal variability [6–10].

The epidemiological studies performed on the different continents revealed remarkable differences between the industrialized and the low-income countries [7, 8, 11–13]. In a thorough review of studies focussing on the distribution of *emm* types between global regions, Steer et *al*. reported that in high- and middle-income countries (Americas, Europe, Asia, and the Middle East), there is essentially a high prevalence of few genotypes (*emm*1, 12, 28, 3, 4, and 89). In contrast, in Africa and the Pacific Islands, the distribution of *emm* types is more diverse and does not show dominant *emm* types [9].

In addition to *emm* genotyping, and based on their tissue tropism, GAS *emm* types can also be grouped into patterns where the patterns A to C strains have a preferential pharyngeal tropism, the pattern D has a cutaneous tropism, and the pattern E which is said to be "generalist" having no specific pharyngeal or cutaneous tropism [14]. In tropical countries, the most frequently isolated *emm* types of GAS belong to *emm*-pattern D (skin tropism) or E (no specific tropism), as opposed to temperate regions where there are more strains of *emm*-pattern A–C (pharyngeal tropism) [15]. The reasons for this contrasting molecular epidemiology are not understood. However, to support the notion that skin *emm* types dominate the epidemiology in many tropical countries, it has been suggested that strains belonging to the pattern E elicit a weaker immune response than throat specialist strains (pattern A-C) [16]. Despite these significant differences in the distribution of *emm* genotypes, regional and temporal differences within industrialized countries remain poorly explained.

Consistent with *emm* typing and *emm* patterns, similar sequences of N terminal part of M proteins, predictable to share functional properties and elicit cross-protective antibodies, has recently been assigned to a specific *emm*-cluster [17]. Therefore, cluster-typing system proposes a new working hypothesis to analyse epidemiological data with a functional and immunological view. From a public health perspective, it could offer the opportunity to understand better the population's immune susceptibility and explain the emergence of new clones, or yet to anticipate a possible vaccination strategy.

Over nine years of the prospective survey, we first aimed to describe clinical and molecular epidemiology of GAS infections in a population of French Brittany. Secondly, hypothesizing that population immunity has to be considered as a risk for clonal emergence or *emm* type

switching, we analysed the dynamic of "Prevalent" and "Emergent" *emm* types by combining *emm* genotyping and *emm*-cluster system.

## Materials and methods

### Study design and case definition

We conducted a prospective study based on the culture-diagnosis of GAS infections from January 1st 2009 to December 31st 2017, at the University Hospital Centre (UHC) of Rennes–France. A case was defined as a patient in whom one or several GAS isolates were collected. For each case, clinical data were collected prospectively by a detailed questionnaire and comprised demographic data (age, sex, residence area...), anatomical site of isolates, clinical presentation (asymptomatic, local signs, fever and/or general signs, hemodynamic shock), clinical diagnosis (as reported in the final medical report for each case), the portal of entry (cutaneous, pharyngeal, anogenital or unknown), risk factors and underlying disease (concomitant surgery, pregnancy, diabetes mellitus, chronic lung and heart failure, intravenous drug abuse, homeless, daily alcohol intake, cirrhosis, steroids medication, solid or haematological malignancy...), and primary treatment management strategy (medical, surgical, requiring or not intensive care). Combined data were validated weekly to identify any inconsistencies and to recover missing data when possible. Cases were classified into three categories:

1. **Carriage**: when clinical symptoms were unrelated to GAS infection.

2. **Non-invasive disease**: when GAS was isolated from non-sterile sites in association with superficial mucosal or cutaneous infections.

3. **Invasive disease**, which is subcategorized in a) **Probable invasive disease**: when GAS was isolated from non-sterile sites but caused an acute illness that required surgery or hospital care, and b) **Definite invasive disease**, when at least one GAS isolate was obtained from a sterile site (e.g., blood, pleural, peritoneal...), or when associated with tissue necrosis or hemodynamic shock, either requiring fluid resuscitation or vasopressor drugs.

The results of the study were reported following the STROBE reporting guidelines for observational studies [18].

### Gas isolates collection

All GAS isolates were collected in the hospital from clinical samples. Most of the isolates have been collected in the University Hospital Centre (UHC) of Rennes (87% of total), where they have been saved exhaustively for nine years, and regardless of infection site or invasiveness. When several isolates were recovered from the same infection case, only the first isolate was then considered to avoid redundancies. The collection was further enriched with GAS isolates sent from other hospital microbiological laboratories of French Brittany: Lorient, Pontivy, and Vannes (11,5% of total recovered isolates), Saint-Brieuc, and Dinan (1,5%). All GAS isolates were identified by Matrix-Assisted Laser Desorption Ionisation–Mass Spectrometry (MALDI-TOF MS, Bruker Daltonics GmbH, Germany). Each isolate was then stored at -80˚C, and sub-cultured at 37˚C with 5% $CO_2$ on Columbia blood agar plates containing 5% sheep blood (Biorad™, France) before performing any experimental procedure.

### Molecular *emm* typing and *emm*-cluster typing system

*Emm* typing was performed by sequencing the 5' portion of the *emm* gene according to the CDC guidelines, and *emm* type was determined by submitting the sequence to CDC *emm* type

database (https://www.cdc.gov/streplab/groupa-strep/index.html; last accessed on 23[th] November 2020). The designation of each *emm*-cluster was then deduced as recently described [17].

### Epidemiological definition

Depending on the strain occurrence observed during the survey period, each *emm* type was classified according to three different dynamic profiles assigned as "Prevalent", "Emergent", and "Sporadic" and defined as follows:

1. "**Prevalent**" ***emm*** **types:** the strains corresponding to genotypes that were isolated continuously, and apart from some fluctuations, their annual rates appeared relatively constant during the study period;

2. "**Emergent**" ***emm*** **types**: the strains corresponding to genotypes that exhibited a sudden change in their incidence, whether this occurred during a specific lag-time or continued over time;

3. "**Sporadic**" ***emm*** **types**: did not correspond to the precedent definitions, and each *emm* type was rarely observed with a prevalence <1% of total isolates.

### Data analysis and statistics

The incidence of invasive infections was estimated using the population statistics of French Brittany regions collected from the National Institute of Statistics and Economic Studies (INSEE, https://www.insee.fr/fr/statistiques/2386251; last accessed on 23[th] November 2020).

Continuous data were expressed as means and standard deviations (SDs), and categorical data as absolute numbers and frequencies. Categorical data were compared by the Chi-square test or Fisher's exact test. The diversity of isolates was expressed by the Simpson's diversity index (SDI) with corresponding 95% confidence intervals ($CI_{95\%}$) calculated using online tools (http://www.comparingpartitions.info/; last accessed on 23[th] November 2020). Logistic regression analyses were conducted to explore the associations of individual risk factors variables (age, sex, comorbidities, or lifestyle risk factors. . .) with invasiveness ("Invasive" versus "Non-invasive") and *emm* dynamic profiles ("Prevalent" versus "Sporadic" and "Prevalent" versus "Emergent"). Variables with a significance level of $p \leq 0.20$ in univariate analyses were included in a multivariate logistic regression model. The values of Odds ratios (OR) and CI95% were adjusted to sex and age. *p*-values $< 0.05$ were considered statistically significant. All statistical analyses were performed using JMP.V13 and SAS[®]V9.4 software (SAS Institute Inc., Cary, USA).

### Ethical statement

Ethical approval, or patients' consent, was not required since the study included only microbiological samples and did not involve human subjects or material. Once validated, the database was completely anonymized.

## Results

### GAS infections and clinical characteristics of the studied population

Between 1st January 2009 and 31st December 2017, GAS isolates were recovered from specimens collected from the skin (38%), oropharyngeal (20%), anogenital (16%), blood (12%),

synovial fluid and bone (6%), pleuro-pulmonary (4%) or other (4%) locations. Several isolates could be collected for a single case, but only the first isolate was attached to each of the 942 recorded cases. The diagnosis of GAS infections was mainly performed within the UHC of Rennes, and explain that, over the 21 regions of French Brittany, the majority of GAS isolates was recovered from patients residing in Rennes and the neighbouring areas (S1 Fig). In the UHC of Rennes, the collection was exhaustive since all the isolated GAS were saved. During the surveillance period, and as previously described, we also observed seasonal variation of the rate of infections (invasive and non-invasive), peaking in autumn/winter (S2 Fig) [19].

Among the 942 identified cases, 49 (5%) were classified as carriage, 889 (94%) as clinical GAS-infection, and only 4 (<1%) were unclassified because of missing clinical data. GAS infections were categorized as non-invasive (350/942, 37%), and invasive infections (539/942; 57%) that were also subcategorized in probable invasive (171/942; 18%), and definite invasive infections (368/942; 39%) (see "Materials and methods" for definitions). Demographic characteristics, clinical features, including main general symptoms, the portal of entry, and positive blood culture rates were reported in Table 1. The median age of the overall population was 31.7 years (ranging from 0 to 102). Age-specific distribution of cases showed three peaks in the [0–5], [30–40], and >70 age-groups (S3 Fig). Focusing on the overall 889 infection cases, the sex ratio (M/F) was 1.16 (54% males); however, the rate of females was over-represented in the [30–40] age-group (64% of females) while the percentage of males was higher in the [40–50] age-group (68% of males) (S3 Fig). While the microbiology laboratory of the UHC of Rennes is the primary laboratory of the area, the exhaustive collection of GAS infections allowed the incidence of invasive infections (probable and definite) to be estimated more accurately for people living in Rennes. Thus, the estimated average of the annual incidence was 5.4 ± 1.3 / 100,000 inhabitant/year with a median of 5.3 and interval of [3.5; 7.3].

As indicated in the final medical report, clinical diagnoses were available for almost all the recorded cases except four and were reported in Fig 1. Proportionally, skin and soft tissue infections were the most frequent (45%), followed by ENT-respiratory (23%), anogenital (15%), and bone and joint infections (10%). Isolated bacteraemia represented 2% of the whole population or 6% of all infections defined as invasive. Notably, in our studied population based on hospital diagnosis, ENT-Respiratory infections were more frequently invasive [non-invasive = 65/350 (19%) vs invasive = 154 / 539 (29%)] and were mainly related to the high proportion of pharyngeal infections that required hospital care or surgical treatment.

The portal of entry was identified for all infection cases except for 31 patients in whom it remained unknown (Table 1) and for whom the final diagnosis was septic arthritis (n = 14), isolated bacteraemia (n = 12), central nervous system infections (n = 2), primary peritonitis (n = 2), and pericarditis (n = 1). Overall the studied population, blood cultures were performed for 462 cases (49%). By focusing on the invasive infections, blood cultures had been performed for 69/172 (40%) of the probable invasive and 293/367 (80%) of definite invasive infections, suggesting that the rate of bacteraemia could be underestimated (Table 1).

Risk factors and comorbidities that could be associated with invasiveness were collected prospectively and reported in Table 1. Remarkably, among all the 889 infection cases, no risk factor or associated comorbidities has been identified for 588 patients (66% of infection cases). By performing univariate analysis of risk factors associated with invasiveness in our studied population, we identified statistically significant associations for age, skin lesion, cardiac failure, surgery <7 days of infection, diabetes, solid cancer and COPD (Table 1). When computed in multivariate logistic regression and adjusted to sex and age, the only independent risk factor remaining in the model and associated with the invasiveness was the age (OR the entire range [CI$_{95\%}$] = 6.35 [3.63, 11.10]; and by unit (year) [CI$_{95\%}$] = 1.018 [1.01, 1.02]; p<0.0001).

**Table 1. Population demography, infection characteristics, and risk factors for invasiveness.**

| | Overall | Carriage | Non-Invasive | Invasive | | *p*-value |
| --- | --- | --- | --- | --- | --- | --- |
| | (n = 942)* | (n = 49) | (n = 350) | **Probable** | **Definite** | **Invasive *vs*** |
| | | | | **(n = 171)** | **(n = 368)** | **Non-invasive** |
| **Age** | | | | | | |
| Mean ± SD | 34.3 ± 26.7 | 34.1 ± 28.3 | 27.0 ± 24.1 | 28.4 ± 22.6 | 43.9 ± 27.7 | <0.0001 |
| Median | 31.7 | 28.5 | 23.8 | 25.3 | 40.4 | |
| [Range] | [0–102] | [0.1–91] | [0–102] | [0.6–92] | [0–97] | |
| **Sex** | | | | | | |
| (% Males) | (52.7) | (33.3) | (50.9) | (59.7) | (53.5) | 0.1911 |
| **Portal of entry; n (% total)** | | | | | | <0.0001 |
| **Cutaneous** | 499 (53) | 5 (10) | 190 (54) | 110 (65) | 191 (52) | |
| **ENT-Respiratory** | 247 (26) | 24 (49) | 64 (18) | 47 (27) | 112 (31) | |
| **Anogenital** | 164 (17) | 20 (41) | 96 (27) | 14 (8) | 34 (9) | |
| **Not Known** | 31 (4) | 0 (0) | 0 (0) | 0 (0) | 31 (8) | |
| *Missing data* | *1* | *-* | *-* | *-* | *-* | |
| **General symptoms related to GAS infection; n (% total)** | | | | | | <0.0001 |
| **None** | 49 (5) | 49 (100) | 0 (0) | 0 (0) | 0 (0) | |
| **Local signs without fever** | 213 (23) | 0 (0) | 176 (50) | 22 (13) | 15 (4) | |
| **Fever and/or sepsis** | 538 (57) | 0 (0) | 148 (42) | 137 (80) | 253 (69) | |
| **Hemodynamic shock** | 88 (9) | 0 (0) | 0 (0) | 0 (0) | 88 (24) | |
| *Missing data* | *54 (6)* | *0 (0)* | *26 (7)* | *12 (7)* | *12 (3)* | |
| **Blood culture; n (% total column)** | | | | | | |
| **Performed** | 462/942 (49) | 13/49(27) | 87/350 (25) | 69/172 (40) | 293/367 (80) | <0.0001 |
| **Positive** | 173/462 (38) | 0/13 (0) | 0/87(0) | 0/69 (0) | 173/293 (59) | - |
| **Risk Factors and associated comorbidities; n (% of total)** | | | | | | |
| **No risk factor** | 613 (65) | 27 (55) | 252 (72) | 121 (71) | 215 (58) | 0.0025 |
| **At least 1 risk factor** | 325 (35) | 22 (45) | 98 (28) | 50 (29) | 153 (42) | 0.0025 |
| **Skin Lesion** | 477 (51) | 8 (16) | 206 (59) | 95 (56) | 166 (45) | 0.0026 |
| **Cardiac Failure** | 34 (4) | 1 (2) | 3 (1) | 4 (2) | 26 (7) | <0.0001 |
| **Surgery <7 days** | 45 (5) | 2 (4) | 9 (3) | 9 (5) | 25 (8) | 0.0105 |
| **Diabetes** | 67 (7) | 2 (4) | 17 (5) | 9 (5) | 39 (11) | 0.0197 |
| **Solid cancer** | 47 (5) | 5 (10) | 10 (3) | 4 (2) | 28 (8) | 0.0282 |
| **COPD** | 14 (1) | 4 (8) | 1 (0.3) | 1 (1) | 8 (2) | 0.0353 |
| **Homeless** | 44 (5) | 2 (4) | 21 (6) | 16 (9) | 4 (1) | 0.1183 |
| **Steroids** | 53 (6) | 7 (14) | 14 (4) | 5(3) | 27 (7) | 0.1923 |
| **IVDU** | 34 (4) | 0 (0) | 11 (3) | 15 (9) | 8 (2) | 0.3829 |
| **Alcohol Abuser** | 59 (6) | 1 (2) | 22 (6) | 16 (9) | 20 (5) | 0.8076 |
| **Blood cancer** | 10 (1) | 0 (0) | 4 (1) | 0 (0) | 6 (2) | 0.9708 |
| **Other Comorbidities** | 132 (14) | 13 (27) | 45 (13) | 20 (12) | 53 (14) | 0.4180 |

The main demographic and clinical data of the studied population were reported according to the given case definitions for "Carriage", "Non-invasive", and "Invasive" (probable and definite invasive) infections. Categorical comparisons between "Invasive" and "Non-invasive" infections were performed with Fisher's exact tests, and *p*-values were indicated in the right column. Abbreviations: COPD = Chronic Obstructive Pulmonary Disease; ENT-Respiratory = Ear-Nose-Throat and respiratory; IVDU = Intravenous drug users.

* Of the 942 recorded cases, 4 had not a clinical diagnosis and several missing clinical data and were not subsequently categorized according to case definition.

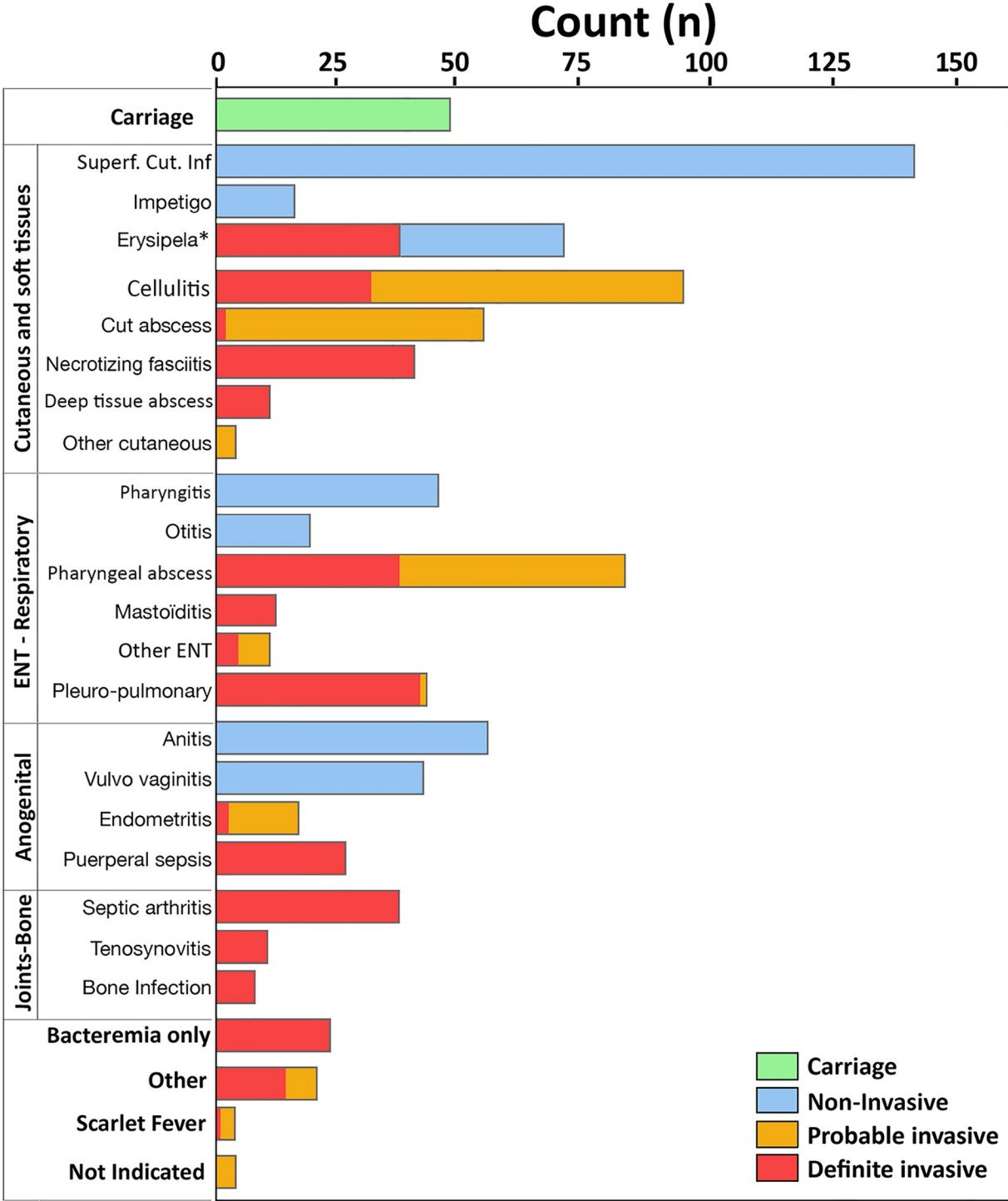

**Fig 1. Clinical diagnosis of GAS infections.** Infection diagnoses were organised according to their main location. For each diagnosis, invasiveness categories ("Carriage", "Non-invasive", "Probable invasive", and "Definite invasive") were indicated by colours (see the legend at the bottom right). *Cases reported as erysipelas were considered as definite invasive infections when blood cultures were positive. The clinical presentations identified as "Other cutaneous" included chronic eschars (n = 3) and cutaneous fistula (n = 1); "Other ENT" included periorbital cellulitis (n = 4), ethmoiditis (n = 2), ocular non-invasive infections (n = 2), pharyngitis with diffuse sinusitis (n = 1), abscess of the nasal septum (n = 1) and peri-tracheal deep infection (n = 1); The group "Other" included peritonitis (n = 4), meningitis/cerebral abscess (n = 3), urinary infection (n = 3), mediastinitis (n = 2), cervical adenitis (n = 1), pacemaker infection (n = 1), pericarditis (n = 1), chorioamnionitis (n = 1) and post-infectious glomerulonephritis (n = 1). For the four patients with unavailable clinical diagnosis, the isolates were collected from a cutaneous specimen. Abbreviations: Cut. abscess = cutaneous abscess; ENT = Ear Nose and Throat; Superf. Cut. Inf = superficial cutaneous infection.

### *Emm*-typing

Molecular *emm*-typing was performed for all the isolates recovered from the 942 recovered cases. We assigned 61 different *emm* types with a Simpson's diversity index (SDI [CI$_{95\%}$]) value = 0.851 [0.779–0.922] (Fig 2 and S1 Table). *Emm* types diversity of GAS isolates collected from cases living "In-Rennes" (SDI [CI$_{95\%}$] = 0.831 [0.777–0.892]), where the collection was almost exhaustive, compared with those collected from other areas grouped in the category "Out-Rennes" (SDI [CI$_{95\%}$] = 0.803 [0.732–0.874]) were not significantly different since their confidence intervals overlapped (S1 Table). Then, all the GAS isolates were grouped and considered as representative for a French Brittany population.

The distribution of *emm* types showed that only four genotypes accounted for more than 50% of all isolated strains namely: *emm*28 (16%), *emm*89 (15%), *emm*1 (14%), and *emm*4 (8%) (Fig 2 and S1 Table). *Emm* types isolated with the highest rate from invasive infections were *emm*3 (89%), *emm*1 (74%), and *emm*87 (74%).

Depending on their occurrence during the survey, each of the 61 identified *emm* types was assigned to one of the three dynamic profiles: "Prevalent", "Sporadic", or "Emergent" (see Materials and methods section for definition). *Emm* types categorized as "Prevalent" encompasses the majority of isolates (n = 686; 72,8%) and corresponded to 9 different *emm* types (*emm* 28, 89, 1, 4, 12, 3, 6, 77, and 2). They were isolated from the beginning of the survey with almost a constant occurrence despite little variations around an individual slope (Fig 2). In

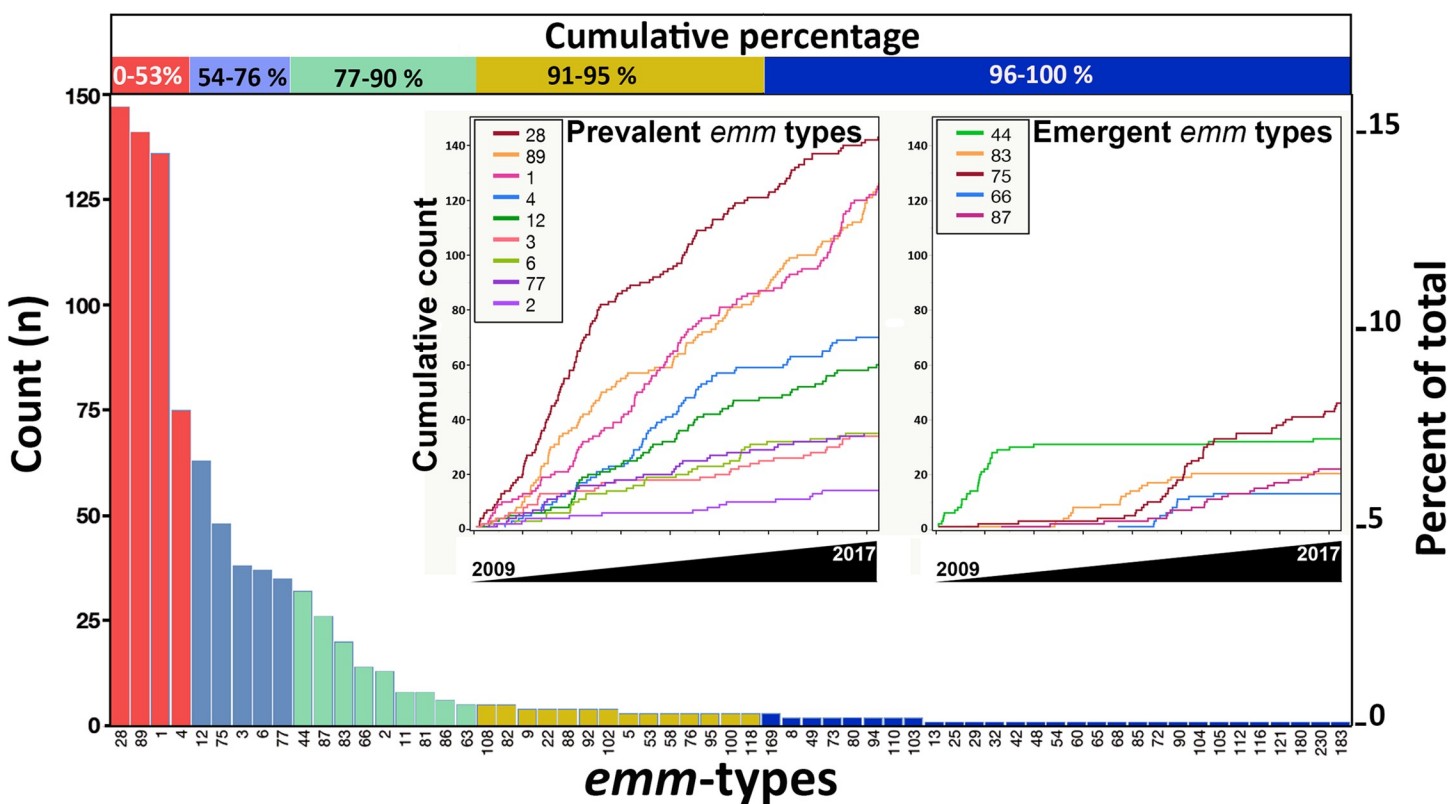

**Fig 2. GAS *emm* types distribution and dynamic profiles.** The distribution of each *emm* type was shown in the main histogram. Cumulative percentages of all the GAS isolated were reported at the top of the figure. Cumulative counts from 2009 to 2017 for "Prevalent" and "Emergent" *emm* types were presented independently. "Prevalent" *emm* types included 9 genotypes (*emm*1, 12, 3, 6, 4 28, 89, 77 and 2), and "Emergent" *emm* types includes 5 different genotypes that emerged successively in the following order: *emm*44 (already present in 2009 and sudden decrease of its occurrence in 2010, quarter (Q)4); *emm*83 (2011; Q2), *emm*75 (2013; Q1), *emm*66 (2013; Q2), and *emm*87 (2013; Q3).

contrast, the majority of *emm* types (47/61 *emm* types) were categorized as "Sporadic", but the isolates represented only 12.4% of all the GAS isolated during the survey, and each of the "Sporadic" *emm* types was rarely isolated ($< 1$% of all the isolates) (S1 Table). "Emergent" *emm* types were marked by a sudden shift in their occurrence during the survey period and corresponded to 5 different genotypes: *emm*44 (n = 32; 3.4% of all the isolates), *emm*66 (n = 14; 1.3%), *emm*75 (n = 48; 4.9%), *emm*83 (n = 20; 2.1%), and *emm*87 (n = 26, 2.7%) (Fig 2).

Depending on the *emm* type identified, each case was assigned to one of the three groups of *emm* type dynamic profiles ("Prevalent", "Sporadic", and "Emergent") that were subsequently analyzed according to the demographic and clinical data, aiming to find specific risk factors that could be associated. As shown in Table 2, the age, the sex, general symptoms related to the infection, the invasiveness of the infection, and the rate of positive blood culture were similar for each of the *emm* type dynamic profiles. The rate of the cutaneous portal of entry was higher for patients infected with "Sporadic" and "Emergent" *emm* types, while ENT-respiratory and anogenital portal of entries were higher when infection occurred with "Prevalent" *emm* types. By performing univariate analysis, risk factors identified to be significantly associated with the group of patients infected with "Emergent" *emm* types were those related to people living in poor hygienic conditions (homeless, alcohol abuse, of IV drug user) (Table 2). By computing the data in a logistic regression model and considering the "Prevalent" *emm* types as a reference category, homeless and alcohol abuse remained both in the model as independent risk factors for the category of patients infected with "Emergent" *emm* types (Table 3). Furthermore, by analyzing risk factors independently for each of the five "Emergent" *emm* types we found that infections with *emm*44 (invasive infections = 16/32; 50%), *emm*66 (7/13; 54%), and *emm*83 (10/20; 50%) were significantly linked to patients living in poor hygienic conditions ($p<0.001$) and associating with one or several risk factors such as homeless, alcoholism or IV drug user. In contrast, we did not find any explanatory clinical risk factors for patients infected with the emergent *emm*75 (invasive infection = 27/48; 56%) and *emm*87 (17/26; 65%) genotypes.

## Dynamic of *emm* types and *emm*-clusters analysis

Protein M is the most immunogenic protein and can confer *emm*-specific immunity against GAS infections. *Emm* types and their distribution were organised according to the recently described cluster classification proposed by Sanderson-Smith [21], that, in addition to the structure and function of the M protein, also consider its capacity to induce an immune cross-protection against the other M proteins belonging to the same cluster. Hypothesizing that most frequent or prevalent *emm* types that circulate in a population may confer a collective immune cross-protection against the other *emm* types from the same cluster, we analysed the relationship of "Emergent" *emm* types to their cluster classification. As represented in Fig 3, the 942 *emm*-typed GAS isolates were assigned to 16 of the 48 described *emm*-clusters and ordered according to their dynamic profile shown by their densities during the survey period (Fig 3A).

"Prevalent" *emm* types (n = 686 GAS isolates; 73%) were classified in *emm*-clusters A-C3 (*emm*1), A-C4 (*emm*12), A-C5 (*emm*3), M6 (*emm*6), E1 (*emm*4), and E4 (*emm*28, *emm*89, *emm*77 and *emm*2).

Almost all the "Sporadic" *emm* types (n = 116 GAS isolates; 12%) belonged to clusters different from those of the "Prevalent" *emm* types except within the cluster E4 (Fig 3B). Remarkably, the *emm*-cluster E4 was the most prevalent and diverse of the other *emm*-clusters and encompassed 11 *emm* types, among which 4 with "Prevalent" (*emm*28, 89, 77, 2) and 7 with "Sporadic" dynamic profiles (*emm*73, 88, 102, 22, 169, 8, 112) (Fig 3B). Notably, the unique *emm*60 identified and categorized as a "Sporadic" *emm* type in the E1 cluster corresponded to

**Table 2. Clinical characteristics, and risk factors for GAS infections with "Emergent" *emm* type: Univariate analyses.**

| Total | Prevalent (n = 686) | Sporadic (n = 116) | Emergent (n = 140) | Univariate p-value |
|---|---|---|---|---|
| **Age** | | | | |
| Mean ± SD | 34.1 ± 27.9 | 36.5 ± 21.9 | 33.3 ± 24.5 | 0.3313 |
| Median | 31.2 | 32.6 | 33.6 | |
| [Range] | [0–97] | [0.4–91] | [0.1–102] | |
| **Sex** | | | | |
| (% Males) | (50.6) | (60.3) | (56.4) | 0.1024 |
| **Portal of entry; n (% total column)** | | | | <0.0001 |
| Cutaneous | 323 (47) | **80 (69)** | **96 (68)** | |
| ENT-Respiratory | **203 (29)** | 19 (16) | 25 (18) | |
| Anogenital | **135 (20)** | 14 (12) | 15 (11) | |
| Not Known | 24 (4) | 3 (3) | 4 (3) | |
| *Missing data* | 1 | 0 | 0 | |
| **General symptoms related to GAS infection; n (% total column)** | | | | 0.0244 |
| None (Carriage) | 43/648 (6) | 1/113 (1) | 5/127 (4) | |
| Local signs without fever | 150/648 (22) | 35/113 (30) | 28/127 (20) | |
| Fever and/or sepsis | 385/648 (56) | 70/113 (62) | 83/127 (59) | |
| Hemodynamic shock | 70/648 (10) | 7/113 (6) | 11/127 (8) | |
| *Missing data* | *38/686 (6)* | *3/116 (3)* | *13/140 (9)* | |
| **Blood culture; n (% total column)** | | | | |
| Performed | 339/686 (49) | 51/116(44) | 71/140 (51) | 0.7489 |
| Positive | 133/339 (39) | 15/51 (29) | 25/71 (35) | 0.6432 |
| **Invasiveness; n (% total column)** | | | | 0.0395 |
| Carriage | 43 (6) | 1 (1) | 5 (3) | |
| Non-Invasive | 242 (35) | 51 (44) | 57 (41) | |
| Probable invasive | 121 (18) | 24 (21) | 26 (19) | |
| Definite invasive | 278 (41) | 39 (34) | 51 (37) | |
| **Risk Factors and associated comorbidities; n (% total cases)** | | | | |
| No-Risk Factor | 461 (67) | 126 (70) | 75 (53) | 0.0057 |
| At least 1 risk factor | 225 (33) | 35 (30) | 65 (47) | 0.0057 |
| Skin Lesion | 321 (47) | 70 (60) | 86(61) | 0.0005 |
| Homeless | 10 (1) | 8 (7) | 26 (19) | <0.0001 |
| Alcohol Abuse | 23 (3) | 6 (5) | 30 (22) | <0.0001 |
| IVDU | 10 (1) | 6 (5) | 18 (13) | <0.0001 |
| Surgery <7 days | 33 (5) | 2 (2) | 5 (6) | 0.1733 |
| Diabetes | 54 (8) | 7 (6) | 6 (4) | 0.2467 |
| Solid cancer | 37 (5) | 3 (3) | 7 (6) | 0.3803 |
| Cardiac Failure | 27 (4) | 2 (2) | 5 (4) | 0.4347 |
| COPD | 12 (2) | 1 (1) | 1 (1) | 0.5029 |
| Steroids | 41 (6) | 5 (4) | 7 (6) | 0.7143 |
| Blood cancer | 8 (1) | 1 (1) | 1 (1) | 0.8616 |
| Other Comorbidities | 103 (15) | 15 (13) | 14 (10) | 0.2563 |

Main clinical data of the studied population were reported according to the given definitions of *emm* dynamic profiles. Univariate categorical comparisons were performed with Fisher's exact tests and the *p*-values were indicated in the right column. Abbreviations: COPD = Chronic Obstructive Pulmonary Disease; ENT-Respiratory = Ear-Nose-Throat and respiratory; IVDU = Intravenous drug users.* Of the 942 recorded cases, 4 had not a clinical diagnosis and several missing clinical data and were not subsequently categorised according to the case definition.

**Table 3. Risk factors for GAS infections with "Emergent" *emm* type: Multinomial logistic regression analyses.**

|  | Prevalent | Sporadic | *p-value* | Emergent | *p-value* |
|---|---|---|---|---|---|
|  | Reference | OR [CI$_{95\%}$] | *(Wald)* | OR [CI$_{95\%}$] | *(Wald)* |
| **Age** (Year) | 1 | 1.00 [0.99–1.01] | 0.4462 | 1.00 [0.99–1.01] | 0.9776 |
| **Sex** (for males) | 1 | 1.32 [0.87–1.99] | 0.1929 | 0.97 [0.66–1.43] | 0.8925 |
| **Cutaneous Lesion** | 1 | **1.63 [1.08–2.46]** | 0.0212 | **1.60 [1.09–2.34]** | 0.0170 |
| **IV Drug User** | 1 | 1.67 [0.45–6.19] | 0.4443 | 1.49 [0.54–4.11] | 0.4429 |
| **Alcohol Abuse** | 1 | 0.28 [0.07–1.12] | 0.0727 | **2.96 [1.47–5.97]** | 0.0024 |
| **Homeless** | 1 | **7.24 [1.78–29.5]** | 0.0058 | **5.58 [2.31–13.5]** | 0.0001 |

"Prevalent" *emm* type served as a reference category for logistic regression. Odds ratios (OR) and CI$_{95\%}$ for risk factors associated with "Sporadic" and "Emergent" *emm* types were estimated by performing two independent logistic regressions adjusted by age and sex.

a patient with a superficial skin infection that occurred during recent touristic travel in Africa (Senegal).

In our study population, "Emergent" *emm* types (n = 140 GAS isolates; 15%) belonged exclusively to clusters D or E, and within which "Sporadic" *emm* types could also be classified. We thus observed sequentially the emergence of the genotypes *emm*44 (before 2009; cluster E3), *emm*83 (2011; cluster D4), *emm*75 (2013; cluster E6), *emm*66 (2013; cluster E2), and *emm*87 (2013; cluster E3). Consistently, after nine years of comprehensive and prospective surveillance in French Brittany, we did not observe any clonal emergence of a new *emm* type within the clusters A-C3, A-C4, A-C5, E1 and E4 that gather "Prevalent" genotypes. Therefore, our observation suggested a complementary hypothesis that "Prevalent" *emm* types would provide a certain degree of immune cross-protection for the population, reducing the probability of allowing the emergence of a new *emm* type within the same cluster. Of note, despite a high diversity of *emm* genotypes found within the cluster E4 (4 "Prevalent" and 7 "Sporadic" *emm*-genotypes), we did not observe the emergence of a new genotype during the study period in this cluster. Finally, *emm* types clustered as a single protein, and for which it has been proposed that their M protein could have different immunological, structural, and functional characteristics were grouped in the same row and encompassed "Prevalent" (*emm*6) and "Sporadic" (*emm*5, *emm*29, and *emm*105) *emm* types (Fig 3).

## Discussion

We presented a comprehensive dynamic of GAS *emm* types over 9-years of prospective culture-based diagnosis in French Brittany. Among the 942 isolates that were clinically documented, 61 different *emm* types were identified. The most "Prevalent" *emm* types were *emm*28, 89, 1, 4, 12, 3, 6, and 77, in agreement with those reported from other studies performed in developed countries [6, 7]. Deciphering the temporal dynamics of the *emm* genotypes in our studied population, we observed that the five "Emergent" *emm* types never belonged to clusters within which "Prevalent" genotypes have been identified.

### Clinical characteristics of the studied population

We initially analysed our population's clinical characteristics, aiming to compare our data with those of other surveys carried out in industrialized countries. Age distribution of GAS infections is generally described with a higher rate in the elderly, followed by infants under 10 years old. This bimodal distribution suggests possible protection by a natural-immunity acquired through multiple episodes of colonization or infection in early life, and that declines in the

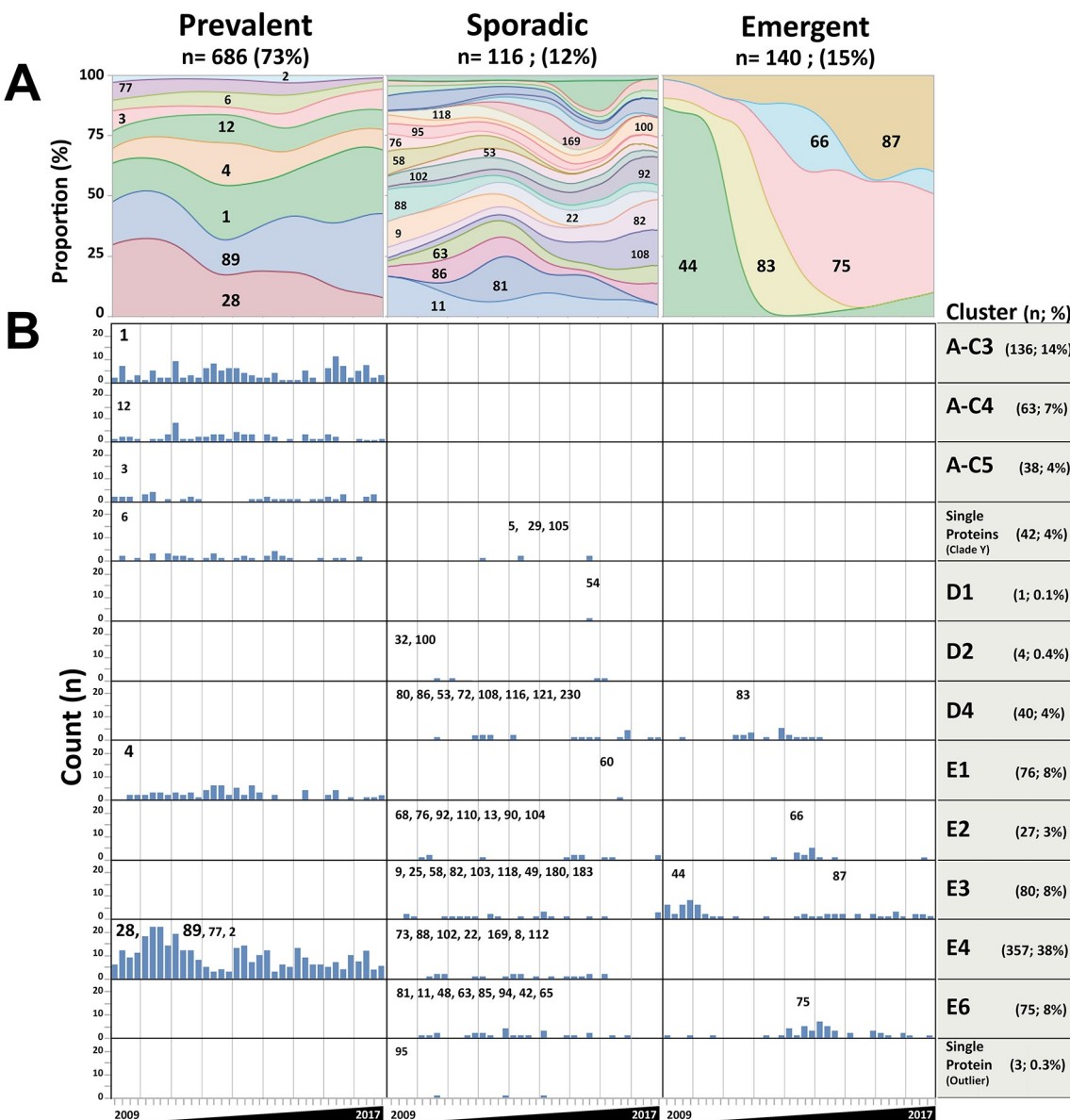

**Fig 3. Dynamic profiles of GAS *emm* types and *emm* cluster-typing classification from 2009 to 2017.** A) The densities of each of the 61 *emm* types were represented and were classified according to their dynamic profile observed during the study period. B) For each *emm* type, the number of isolates from 2009 to 2017 was reported by quarter. They were organised according to their dynamic profile (column) and their corresponding cluster (row, indicated at the right side). *Emm* types were indicated in the corresponding box. E*mm* types 6, 5, 29 and 105 were grouped within a single row named "Single protein (clade Y)", and corresponded to four individual clusters, M6, M5, M29 and M105 respectively. "Prevalent" *emm* types (*emm*1, 12, 3, 6, 4 28, 89, 77 and 2) belonged to the clusters A-C3 to 5, M6, E1, and E4. "Emergent" *emm* types (*emm*44, 66, 82, 75, and 87) belonged to clusters D4, E2 and E3 that were different from those of the "Prevalent" *emm* types.

elderly [20, 21]. As reported by Lamagni *et al* [22], we have also observed increased infection rate between 20 and 40 years. Regarding the UK population, it has been proposed that this reshaping of age distribution results from a high rate of intravenous drug users [22]. In the same way of evidence, we noticed for this age group category an excess of patients having one or several risk factors such as homeless, alcoholism, or intravenous drug use. Besides, we have also observed a high genital tract sepsis rate in women of childbearing age (between 30 and 40

years old). The primary diagnoses were endometritis and postpartum puerperal sepsis that accounted for about 20% of all the invasive infections for this age category. It has been suggested that altered immune status during pregnancy and specific characteristics of the infecting GAS strain contribute to the risk for GAS infection and mortality in postpartum women [23].

For the overall population, risk factors for invasiveness identified by univariate analysis (age, diabetes, cardiac failure, and malignancy) are consistent with other studies performed in industrialized countries [6, 7, 19, 24]. However, the rate of risk factors identified above increased with ageing, and when analysed with a multivariate logistic regression model, the age remained the unique independent risk factor in our studied population. Gender as a risk factor varies between studies, and this is possibly dependent on the age group distribution of each studied population [7, 13, 22, 25–28].

Skin and soft tissue infections were the most frequent clinical presentations, and the severity of the infections required hospitalization for the majority of them. Among cases diagnosed as erysipelas, although frequently described as restricted to the superficial skin and considered as a non-invasive infection, 48% of them have a positive blood culture and subsequently categorised as invasive infections. This may indicate that clinical differentiation of erysipelas is not precise enough, and streptococcal cellulitis could be underdiagnosed.

Perianal streptococcal dermatitis or anitis is the most common non-invasive disease seen in children of 3–5 years old [29]. The four most frequent types found for this infection were *emm*28 (56%), *emm*77 (9%), *emm*4 (13%), and *emm*12 (6%). Of note *emm*28 an *emm*77 express the protein R28, which has an LPxTG motif ([30, 31] and personal data) and believed to promote adhesion to human cervical cells.

## Dynamic of *emm* types

For several decades, it has been known that the most potent protective immunity against GAS infection is M specific [3], which produces opsonizing antibodies directed against the N terminus of the M protein. Molecular types sharing structural and functional homologies were inferred to a unique *emm*-cluster and could elicit cross-protective immunity of almost all *emm* types within a specific *emm*-cluster [32]. As in all the surveys performed in industrialized countries, the throat specialist genotypes *emm*1 (belonging to cluster A-C3), *emm*12 (A-C4), and *emm*3 (A-C5) are "Prevalent" [12, 33, 34], and they are characterized by their ability to have fibrinogen binding properties accounting for a high rate of invasive manifestations. Throughout the nine years of our surveillance, the genotypes *emm*1, 3 and 12, were dominant *emm* types without any other *emm* type competitors identified within their specific *emm*-cluster. Their epidemiological dominance and persistence are not well understood and could be explained by the absence of other circulating *emm* types belonging to their specific *emm*-clusters. An alternative explanation could be a complex antigenic structure or a specific dynamic for genetic evolution affecting immunogenic epitopes of many A–C *emm* types with throat tropism [35], preventing a stable and highly-specific long-lasting immunity.

The high prevalence and diversity of *emm* types encountered for the cluster E4 corroborate other studies [8, 33], and may indicate a variable or insufficient cluster-specific natural immune cross-protection. Recent work investigating the cross-protection capabilities against the 17 *emm* types of the cluster E4 identified the potential requirement of five M peptides (*emm*2, 8, 22, 89, and 112) to induce a bactericidal cross activity against 15/17 E4 GAS, excepting *emm*77 and *emm*114 [32]. Notably, we never recorded any "Emergent" *emm* type for this cluster, but seven "Sporadic" *emm*-genotypes were identified.

"Emergent" *emm* types occurred as an epidemiological shift within the clusters D4, E2, E3 and E6 that were free of any "Prevalent" *emm* types during all the study period. Mechanisms

that can contribute to the emergence of one or more genotypes in a population are not well understood. However, risk factors and genetic modifications of the strain, including the acquisition of new virulence factors, may play variable roles depending on the *emm* type. Although more challenging to assess, another complementary factor is the lack of protective immunity of the population against GAS, which can thus facilitate the emergence of a specific *emm* type. Most of *emm*44, 83 and 66 strains were isolated from patients with specific risk factors such as living in poor conditions and big cities. As we reported previously, the whole genome analysis of some of emergent strains in French Brittany identified a genetic acquisition of new transposons for *emm*44, and *emm*83, and mutations resulting in a null allele of a stand-alone RopB regulator for *emm*66 [36–39]. The role of these genetic modifications as an explanatory mechanism for clonal emergence remained unknown, and the increase in infection incidence was recorded for only 2 to 3 years. These observations are consistent with studies reporting that short adaptive evolution driven by habitat adaptation (skin or generalist rather than throat specialist strains) underwent horizontal gene transfer events that could offer selective advantages in a susceptible population, either lacking immune protection or having a specific risk factor [35] as we observed in our population.

The European survey published in 2009 indicated that infections with *emm*75 strains were found only in few countries (Finland, Greece, Germany, and Romania), but remained marginal among the "Prevalent" *emm* types [6]. As we previously reported [40], the sharp increase of *emm*75 infection rate observed in 2013 was most likely related to the emergence of a new clone that acquired two new prophages encoding virulence factors (SpeC and SpeK superantigens). Herein, we failed to identify any specific risk factor (clinical or behavioural) that could explain the emergence of the genotype *emm*75 in a susceptible population. However, the genotype *emm*75 tended to become prevalent in the French Brittany population where it represents 4 to 6% of strains isolated annually. We do not know if the sustained rise of the *emm*75 genotype will continue, or if we will observe upsurges or epidemic waves in French Brittany as in other geographic regions. In our opinion, the emergence of genotype *emm*75 needs careful consideration. First, an *emm*75 strain isolated from blood culture in 2015 in the UK and recently sequenced (Strain: NCTC13751, GeneBank accession: LS483437) exhibits the same genetic modifications that we have observed in strains isolated in French Brittany. Second, it has recently been reported in Portugal an increasing trend of invasive infections due to the genotype *emm*75 that also shares the superantigens genes *speC* and *speK* [41]. All these strains deserve to be analyzed more in-depth to decipher if this emergence corresponded to the same clonal spread. Our observation can be paralleled with the nationwide increase in invasive disease due to the genotype *emm*89. This genotype upsurges last decades and has recently been associated with the emergence of a new successful clade variant that has undergone several genetic modifications affecting known virulence factors [42].

Finally, the emergence of the *emm*87 genotype observed in 2013 is remarkable because it predominates in England while seldom isolated in the rest of Europe [6]. The spreading of the genotype *emm*87 may have occurred in French Brittany, given the geographical proximity and frequent exchanges between the two countries.

The monocentric design is the main limitation of our study, and other population-based investigations are required to confirm our findings. However, many strengths have to be considered, including the prospective and longitudinal collection of strains from invasive and non-invasive infections with their attached clinical data. Also, the geographic delimitation to a population-based recovery of GAS strain enabled us to observe a comprehensive dynamic of circulating *emm* types.

After nine years of GAS infection surveillance, we described a high diversity of circulating GAS *emm* types and characterized accurately epidemiological shifts and dynamic profiles of

five successive "Emergent" *emm* types *(emm*44, 66, 75, 83 and 87). They occurred within *emm*-clusters different from those gathering "Prevalent" *emm* types that could suggest a population susceptibility potentially due to a weak natural immune cluster-specific cross-protection. The emergence of the genotype *emm*75 occurred in 2013 is now marked by a sustained prevalence suggesting a potential expansion of a successful clone. Dynamic monitoring of GAS infections by combining at least molecular *emm* typing and cluster classification remains the keystone strategy for epidemiological surveillance.

## Supporting information

**S1 Fig. Geographical distribution of collected cases in French Brittany according to the residence of patients.** The reported numbers corresponded to the 21 areas of French Brittany: 1) Auray; 2) Brest; 3) Broceliande; 4) Centre Bretagne; 5) COB; 6) Cornouaille; 7) Dinan; 8) Fougères; 9) Guingamp; 10) Lorient; 11) Morlaix; 12) Ploermel; 13) Pontivy; 14) Redon; 15) Rennes; 16) Saint-Malo; 17) Saint Brieuc; 18) Tregor-Goelo; 19) Vallons-Vilaine; 20) Vannes; 21) Vitré. Regional hospitals (*) and University Hospital Centres (•) were indicated on the map. For each area, the average number of cases collected/100,000 inhabitants/year were reported according to the colour legend.
(DOCX)

**S2 Fig. Seasonal variation of infection rates.** All infections (green), non-invasive (blue), and invasive infections (red) were broken down by year, and rates of infections were given for each quarter. 1: January to March; 2: April to June; 3: July to September; 4: October to December.
(DOCX)

**S3 Fig. Age group distribution of GAS infections.** The rates for males, invasive infections and the portal of entry were indicated for each age group. ENT-Resp = Ear Nose and Throat and Respiratory. *n = 889 infections/942 collected cases (49 carriage and 4 cases with missing values were not included).
(DOCX)

**S1 Table.** *Emm* **types diversity "In Rennes" area and "Out Rennes" grouped areas.** For each identified *emm* types, the total number of GAS isolates (n) and percentage of the total (%) were indicated in the corresponding column. For the most frequent genotypes (n > 10 isolates) we performed a categorical analysis (Fisher's exact test) to compare the rate of their occurrence "In Rennes" vs "Out Rennes" groups. Simpson's Indexes of Diversity (SDI) and their comparison were given at the bottom of the table. * Among the 942 *emm*-typed GAS isolates, 1 missed value for the residential area.
(DOCX)

**S1 File.**
(XLSX)

## Acknowledgments

We would like to thank Dr Pascal Vincent (microbiologist in the UHC of Rennes and currently retired) and Dr Jean-Francois Ygout (microbiologist in the general Hospital of Lorient and currently retired) for their contribution to this work by managing the database and sending GAS isolates, respectively, the microbiologists of the hospital of Dinan, Lorient, Pontivy, Saint-Brieuc, and Vannes for sending GAS isolates, all the members of the Department of Bacteriology—UHC of Rennes for their technical support and assistance in this study.

## Author Contributions

**Conceptualization:** Ahmad Faili, Samer Kayal.

**Data curation:** Sarrah Boukthir, Séverine Moullec, Jeff Morcet, Ahmad Faili, Samer Kayal.

**Formal analysis:** Sarrah Boukthir, Séverine Moullec, Alexandra Meygret, Jeff Morcet, Ahmad Faili, Samer Kayal.

**Investigation:** Sarrah Boukthir, Séverine Moullec, Marie-Estelle Cariou, Alexandra Meygret, Ahmad Faili, Samer Kayal.

**Methodology:** Séverine Moullec, Ahmad Faili, Samer Kayal.

**Project administration:** Ahmad Faili, Samer Kayal.

**Supervision:** Ahmad Faili, Samer Kayal.

**Validation:** Sarrah Boukthir, Séverine Moullec, Ahmad Faili, Samer Kayal.

**Writing – original draft:** Sarrah Boukthir.

**Writing – review & editing:** Samer Kayal.

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
