## [Decision Letter · Decision Letter 0]

17 Nov 2020

PONE-D-20-31664

A prospective survey of Streptococcus pyogenes infections in French Brittany from 2009 to 2017: Comprehensive dynamic of new emergent emm genotypes.

PLOS ONE

Dear Dr. Kayal,

Thank you for submitting your manuscript to PLOS ONE. After careful consideration, we feel that it has merit but does not fully meet PLOS ONE’s publication criteria as it currently stands. Therefore, we invite you to submit a revised version of the manuscript that addresses the points raised during the review process.

The manuscript has been assessed by two reviewers. Their comments are available below. The reviewers have raised a number of concerns about the the data, they recommend revisions to provide a fuller outline of the methodology and main results. Please carefully revise the manuscript to address all the points raised by the two reviewers.

We look forward to receiving your revised manuscript.

Kind regards,

Jose Melo-Cristino, M.D., Ph.D.

Academic Editor

PLOS ONE

Journal Requirements:

Reviewers' comments:

Reviewer's Responses to Questions

**Comments to the Author**

1. Is the manuscript technically sound, and do the data support the conclusions?

Reviewer #1: Yes

Reviewer #2: Yes

2. Has the statistical analysis been performed appropriately and rigorously? 

Reviewer #1: Yes

Reviewer #2: I Don't Know

3. Have the authors made all data underlying the findings in their manuscript fully available?

Reviewer #1: Yes

Reviewer #2: Yes

4. Is the manuscript presented in an intelligible fashion and written in standard English?

Reviewer #1: Yes

Reviewer #2: Yes

5. Review Comments to the Author

Reviewer #1: Overall this is an interesting study describing the clinical and genotying features of GAS infections in French Brittany between 2009 and 2017. The authors highlight the emergence of new emm-types and their association with clusters that differ from the prevalent emm types.

Comments

Abstract Line 22 – think there needs to be an additional word here such as ‘complex epidemiology driven by the diversity’

Abstract line 26 – I don’t think this study has the power to decipher the underlying mechanism for the emergence of successful clones given that only emm typing was performed.

Introduction

Line 61 – delete the word species

Methods

Line 130 – I am not quite clear on what is meant by collected exhaustively. Are all isolates routinely saved regardless of infection site? Is there likely to be an over-representation of invasive or severe GAS isolates? How much non-invasive or carriage would be collected?

Results

Line 177 – is the number of cases that same as the number of isolates or were several isolates obtained from the same patient?

Line 192 (and other places) – two open square brackets are used rather than an open and close.

Line 195 – what is meant here? That non-invasive infections would not be recorded or isolates not saved? Or is this something to do with management – ie no clinical outcomes or information recorded. I think a bit of clarity here (or in the Methods) would be good to try and get an idea of what sort of isolates would be included – it is not clear what sort of number of non-invasive infections would be recorded/collected. Also relates to Line 218-220 – is this related to the types on non-invasive isolates obtained here and only serious ones potentially related to invasive disease would be collected?

Lines 243-245 – this section is a bit confusing. Is it 51% of invasive cases? Then 49% would not and this can’t be ‘most’. In the sentence starting ‘Its proportion’ what does the ‘it’ refer to?

Line 255 – delete the word ‘finally’

Line 262 – ‘showed’ rather than ‘show’

Line 281 – ‘in their occurrence’ rather than ‘of their occurrence’

Line 301 – ‘associated with’ rather than ‘associating one’

Lines 321-325 should be in the Discussion

Line 332 – hypothesising

Line 363 – I do not really understand this sentence. Do you mean that no emergent emm types belonged to A-C, which was the most common of the prevalent genotypes?

Line 378 – an extra bracket here. Also change ‘agree closely’ to ‘similar to’ or ‘in agreement with’

Line 380 – suggest changing to ‘that belonged to emm-clusters that “Sporadic” but not “Prevalent” genotypes were members of’ to improve clarity.

Line 388 – delete the ‘the’ after observed

Line 398 – delete the ‘the’ before invasiveness

Line 400- what is meant here? The number of risk factors increase with age?

Line 402 – delete the ‘The’ before gender

Line 413 – emm28 and ee77 have R28 but not 4 and 12.

Discussion

Line 416 – ‘it has been known’ rather than ‘admitted’

Line 424 – ‘were dominant’ rather than ‘behaved lie’

Line 433 – delete ‘A’ before recent

Line 439 – not sure what is meant here? No prevalent emm types were recorded during this time or that no prevalent emm types were of clusters D and E?

Line 400 – ‘Were’ rather than ‘Have’ and ‘patients with’ rather than ‘patients having’

Line 455 – ‘was’ rather than ‘has been recorded’

Line 457 – should it be ‘where it represents’ rather than ‘while it represents’?

Line 458 – do you mean if emm75 will become a “Prevalent” emm type?

Line 472 – ‘may’ rather than ‘probably’

Line 483 – ‘potentially’ rather than ‘presumably’

Tables – it is not standard practice to have legends for Tables. A brief title is included and then any parts that need clarification should be footnotes. I don’t think there should be a part A and B to table 2 – this should be Table 2 and Table 3.

Throughout the Methods and some of the Results the term ‘Strain’ is used when it should be ‘Isolate’

Reviewer #2: Boukthir et al report on a prospective survey of Streptococcus pyogenes infections in French Brittany from 2009 to 2017. It is an excellent study with good data. I can fully recommend the publication and have only a few minor comments. A small restriction from my side: I cannot judge the statistics sufficiently well.

minor remarks

lines 192-195 (and S3 Fig)

To me the arrangement of the brackets for the representation of the age intervals seems to take some getting used to.

lines 194-195

The percentages of 65% and 63% do not correspond to those given in S3 Figure. Furthermore, table 1 does not differentiate the different age intervals at all.

table 1

Portal of Entry, Overall: 499, 247, 164 and 31 sum up (only) to 941 (not 942)?

lines 246-247

Where does the number of 588 cases come from? I cannot find it anywhere else in the manuscript.

Furthermore: 588/889 = 0.661417322… (=> 66%, not 67% as indicated)

line 262

shows (not shown)

line 263

The distribution can be seen better in table S1 than in figure 2.

line 300

7/13 = 0.538461538 => (=> 54%, not 53% as indicated)

lines 329-331

Please rephrase.

lines 334-335

figure 3 (Fig 3) => Please remove the duplication.

lines 339-347

I am not sure if I understand the description correctly, especially the front part. Probably this is fine, but if necessary it could be made a bit clearer.

lines 377-378

The order of the emm types mentioned is different from that shown in figure 2 and table S1. Is this intentional?

line 442

transposons => please correct

line 455

…a specific… factors… => please rephrase

line 516

Ygout (?)

figure 2

It would be nice if the percent sign could also be included at 0-53 and 54-76 (top left).

figure 3

It would be nice if in the middle column with the sporadic infections in figure 3a some more emm types could be marked in the figure.

S3 figure

To me the arrangement of the brackets for the representation of the age intervals seems to take some getting used to (see also lines 192-195).

The percentages of 65% and 63% given in lines 194-195 do not correspond to those given in S3 Figure.

For the age group of [30-40[, there should be a ‘-‘ instead of a ‘;’.

6. PLOS authors have the option to publish the peer review history of their article (what does this mean?). If published, this will include your full peer review and any attached files.

Reviewer #1: No

Reviewer #2: No

---

## [Author Response · Author response to Decision Letter 0]

23 Nov 2020

Response to reviewers

Reviewers' comments:

Reviewer's Responses to Questions

Comments to the Author

1. Is the manuscript technically sound, and do the data support the conclusions?

Reviewer #1: Yes

Reviewer #2: Yes

2. Has the statistical analysis been performed appropriately and rigorously? 

Reviewer #1: Yes

Reviewer #2: I Don't Know

We wish to indicate that all the statistics have been carried out and reviewed by Jeff Morcet, who is a statistician and signatory of the article.

3. Have the authors made all data underlying the findings in their manuscript fully available?

Reviewer #1: Yes

Reviewer #2: Yes

4. Is the manuscript presented in an intelligible fashion and written in standard English?

Reviewer #1: Yes

Reviewer #2: Yes

5. Review Comments to the Author

Reviewer #1: Overall this is an interesting study describing the clinical and genotying features of GAS infections in French Brittany between 2009 and 2017. The authors highlight the emergence of new emm-types and their association with clusters that differ from the prevalent emm types.

Comments

Abstract Line 22 – think there needs to be an additional word here such as ‘complex epidemiology driven by the diversity’

As suggested we added the word driven as follows:

”…complex epidemiology driven by the diversity, …”

Abstract line 26 – I don’t think this study has the power to decipher the underlying mechanism for the emergence of successful clones given that only emm typing was performed.

The sentence has been modified by the follows: 

“This study, based on a longitudinal population survey, aimed to understand the dynamic of GAS emm types and to give leads to better appreciate underlying mechanisms for the emergence of successful clones. »

Introduction

Line 61 – delete the word species

The word species has been deleted.

Methods

Line 130 – I am not quite clear on what is meant by collected exhaustively. Are all isolates routinely saved regardless of infection site? Is there likely to be an over-representation of invasive or severe GAS isolates? How much non-invasive or carriage would be collected?

All the strains isolated routinely in the hospital laboratory were collected and the clinal data were saved regardless of infection site or invasiveness, and when several strains were isolated for a patient only the first strain was then considered. In the revised version, this has been stated as follows: 

“All GAS isolates were collected in the hospital from clinical samples. Most of the isolates have been collected in the University Hospital Centre (UHC) of Rennes (87% of total), where they have been saved exhaustively for nine years, and regardless of infection site or invasiveness. When several isolates were recovered from the same infection case, only the first isolate was then considered to avoid redundancies.”

Results

Line 177 – is the number of cases that same as the number of isolates or were several isolates obtained from the same patient?

The sentence has been modified as follows: 

“Between 1st January 2009 and 31st December 2017, GAS isolates were recovered from specimens collected from the skin (38%), oropharyngeal (20%), anogenital (16%), blood (12%), synovial fluid and bone (6%), pleuro-pulmonary (4%) or other (4%) locations. Several isolates could be collected for a single case, but only the first isolate was attached to each of the 942 recorded cases.”

Line 192 (and other places) – two open square brackets are used rather than an open and close.

We have used open brackets to indicate intervals. We corrected all the second open brackets by the following notation [a,b) to indicate an interval from “a” to “b” that is inclusive of “a” but exclusive of “b”. The corrections have been made in the manuscript and supporting figures S1 and S3. 

Line 195 – what is meant here? That non-invasive infections would not be recorded or isolates not saved? Or is this something to do with management – ie no clinical outcomes or information recorded. I think a bit of clarity here (or in the Methods) would be good to try and get an idea of what sort of isolates would be included – it is not clear what sort of number of non-invasive infections would be recorded/collected. Also relates to Line 218-220 – is this related to the types on non-invasive isolates obtained here and only serious ones potentially related to invasive disease would be collected?

We systematically saved all the GAS isolated routinely in our laboratory and recorded and analysed all the corresponding infection cases regardless of their invasiveness. However, in order to estimate the incidence of GAS infections, we felt that it was necessary to indicate that in our studied population (based on hospital diagnosis) the incidence of non-invasive infections could not be estimated accurately since most of the patients with a non-invasive infection do not require any hospital care. Thus, we were only able to estimate the incidence of invasive infections, and more accurately for the cases living in Rennes where we believe that almost all the patients with an invasive infection came to the hospital. 

For more clarity, the sentence starting on Line 195-197(first version) has been deleted since it does not add any meaningful information to the estimation of the incidence of invasive infections. 

Line 218-220. Insofar, as the diagnoses are exclusively performed in the hospital, we believe that pharyngitis, which is the most frequent of GAS infections, is managed of by out-of-hospital medicine, mostly without a bacteriological diagnosis, and therefore not collected during our survey. We have clarified this aspect by modifying the sentence as follows.: 

“Notably, in our studied population based on hospital diagnosis, ENT-Respiratory infections were more frequently invasive [non-invasive = 65/350 (19%) vs invasive = 154 / 539 (29%)] and were mainly related to the high proportion of pharyngeal infections that required hospital care or surgical treatment.”

Lines 243-245 – this section is a bit confusing. Is it 51% of invasive cases? Then 49% would not and this can’t be ‘most’. In the sentence starting ‘Its proportion’ what does the ‘it’ refer to?

We agree with the reviewer that this section could be a bit confusing, and it seemed suitable to us to delete these two phrases (from Line 242 to line 246 in the first version) to allow a more comfortable reading of this paragraph that focused on the analysis of risk factors for invasive infections.

Line 255 – delete the word ‘finally’

The word “finally” has been deleted. 

Line 262 – ‘showed’ rather than ‘show’

The modification has been made. 

Line 281 – ‘in their occurrence’ rather than ‘of their occurrence’

The modification has been made. 

Line 301 – ‘associated with’ rather than ‘associating one’

The modification has been made. 

Lines 321-325 should be in the Discussion

As suggested, this paragraph has been moved to the discussion section.

Line 332 – hypothesizing

The correction has been made. 

Line 363 – I do not really understand this sentence. Do you mean that no emergent emm types belonged to A-C, which was the most common of the prevalent genotypes?

We have changed the sentence as follows

“Consistently, after nine years of comprehensive and prospective surveillance in French Brittany, we did not observe any clonal emergence of a new emm type within the clusters A-C3, A-C4, A-C5, E1 and E4 that gather “Prevalent” genotypes.

Line 378 – an extra bracket here. Also change ‘agree closely’ to ‘similar to’ or ‘in agreement with’

Correction and suggested modification have been made. 

Line 380 – suggest changing to ‘that belonged to emm-clusters that “Sporadic” but not “Prevalent” genotypes were members of’ to improve clarity.

We changed the sentence as suggested.

Line 388 – delete the ‘the’ after observed

This has been deleted.

Line 398 – delete the ‘the’ before invasiveness

This has been deleted.

Line 400- what is meant here? The number of risk factors increase with age?

The sentence has been changed as follows:

“However, the rate of risk factors identified above increased with ageing and when analysed with a multivariate logistic regression model….”

Line 402 – delete the ‘The’ before gender

This has been deleted

Line 413 – emm28 and ee77 have R28 but not 4 and 12.

In the new version of the manuscript this has been specified as follows: 

“The four most frequent types found for this infection were emm28 (56%), emm77 (9%), emm4 (13%), and emm12 (6%). Of note emm28 and emm77 express the protein R28, which has an LPxTG motif ([34, 35] and personal data) and believed to promote adhesion to human cervical cells.”

Discussion

Line 416 – ‘it has been known’ rather than ‘admitted’

The modification has been made. 

Line 424 – ‘were dominant’ rather than ‘behaved lie’

The modification has been made. 

Line 433 – delete ‘A’ before recent

The modification has been made. 

Line 439 – not sure what is meant here? No prevalent emm types were recorded during this time or that no prevalent emm types were of clusters D and E?

To be more understandable we have better specified the clusters concerned by the emergent emm types: 

““Emergent” emm types occurred as an epidemiological shift within the clusters D4, E2, E3 and E6, that were free of any “Prevalent” emm types during all the study period.”

Line 400 – ‘Were’ rather than ‘Have’ and ‘patients with’ rather than ‘patients having’

Modifications have been made (line 440).

Line 455 – ‘was’ rather than ‘has been recorded’

The modification has been made (line 445).

Line 457 – should it be ‘where it represents’ rather than ‘while it represents’?

The modification has been made. 

Line 458 – do you mean if emm75 will become a “Prevalent” emm type?

Actually, we cannot assert that the incidence of the emergent emm75 will continue to remain constant as we observed for the dynamic “Prevalent” emm types. Then to avoid misinterpretation, we have modified the sentence as follows

“We do not know if the sustained rise of the emm75 genotype will continue or if we will observe upsurges or epidemic waves in French Brittany as in other geographic regions.”

Line 472 – ‘may’ rather than ‘probably’

The modification of the sentence has been done as follows: 

“The spreading of the genotype emm87 may have occurred in French Brittany, given the geographical proximity and frequent exchanges between the two countries”

Line 483 – ‘potentially’ rather than ‘presumably’

The modification has been made. 

Tables – it is not standard practice to have legends for Tables. A brief title is included and then any parts that need clarification should be footnotes. I don’t think there should be a part A and B to table 2 – this should be Table 2 and Table 3.

We have followed all the suggestions and split the table 2 (A and B) in two different tables; Table 2 and Table 3. The reference for table 2 and 3 has also been modified accordingly in the text.

Throughout the Methods and some of the Results the term ‘Strain’ is used when it should be ‘Isolate’

Throughout the text we have changed “strain” to “isolate” when necessary. All the modifications have been highlighted in the revised version of the manuscript. 

-

Reviewer #2: Boukthir et al report on a prospective survey of Streptococcus pyogenes infections in French Brittany from 2009 to 2017. It is an excellent study with good data. I can fully recommend the publication and have only a few minor comments. A small restriction from my side: I cannot judge the statistics sufficiently well.

minor remarks

lines 192-195 (and S3 Fig)

To me the arrangement of the brackets for the representation of the age intervals seems to take some getting used to.

We have used open brackets to indicate intervals. We corrected all the second open brackets by the following notation [a,b) to indicate an interval from “a” to “b” that is inclusive of a but exclusive of “b”. The corrections have been made in the manuscript and supporting figures S1 and S3. 

lines 194-195

The percentages of 65% and 63% do not correspond to those given in S3 Figure. Furthermore, table 1 does not differentiate the different age intervals at all.

The sentence has been modified. See bellow the comment concerning S3 Figure.

table 1

Portal of Entry, Overall: 499, 247, 164 and 31 sum up (only) to 941 (not 942)?

Among the 4 patients for whom several clinical information were missing as indicated at the top of the column “Overall (942)*”, the portal of entry is still missing for one of them. Then we have added a line to the table 1 and table 2 to indicate this missing data in the corresponding column.

lines 246-247

Where does the number of 588 cases come from? I cannot find it anywhere else in the manuscript.

Furthermore: 588/889 = 0.661417322… (=> 66%, not 67% as indicated)

For more clarity, the sentence “Remarkably, most of the infection cases (588/889, 67%) had no identified risk factors or associated comorbidities.” has been modified as follows:

“Remarkably, among all the 889 infection cases, no risk factor or associated comorbidities has been identified for 588 patients (66% of infection cases).”

line 262

shows (not shown)

The modification has been made according to the suggestion of the reviewer #1 

line 263

The distribution can be seen better in table S1 than in figure 2.

As suggested the reference to table S1 was also added as follows: 

“… and emm4 (8%) (Fig 2 and Table S1).” 

line 300

7/13 = 0.538461538 => (=> 54%, not 53% as indicated)

The correction has been made.

lines 329-331

Please rephrase.

We have rephrased the entire sentence as follows:

“Emm types and their distribution were organised according to the recently described cluster classification proposed by Sanderson-Smith [21], that, in addition to the structure and function of the M protein, also consider its capacity to induce an immune cross-protection against the other M proteins belonging to the same cluster.”

lines 334-335

figure 3 (Fig 3) => Please remove the duplication.

The duplication has been removed.

lines 339-347

I am not sure if I understand the description correctly, especially the front part. Probably this is fine, but if necessary it could be made a bit clearer.

The description has been slightly modified as follows: 

“Fig 3. Dynamic profiles of GAS emm types and emm cluster-typing classification from 2009 to 2017. A) The densities of each of the 61 emm types were represented and were classified according to their dynamic profile observed during the study period. B) For each emm type, the number of isolates from 2009 to 2017 was reported by quarter. They were organised according to their dynamic profile (column) and their corresponding cluster (row, indicated at the right side). Emm types were indicated in the corresponding box. Emm types 6, 5, 29 and 105 were grouped within a single row named “Single protein (clade Y)”, and corresponded to four individual clusters, M6, M5, M29 and M105 respectively. “Prevalent” emm types (emm1, 12, 3, 6, 4 28, 89, 77 and 2) belonged to the clusters A-C3 to 5, M6, E1, and E4. “Emergent” emm types (emm44, 66, 82, 75, and 87) belonged to clusters D4, E2 and E3 that were different from those of the “Prevalent” emm types.”

There were also errors in the number of isolates indicated in the upper part of the figure, and for each of the dynamic profiles. They have been corrected.

lines 377-378

The order of the emm types mentioned is different from that shown in figure 2 and table S1. Is this intentional?

Emm types indicated in the text have been reordered according to the figure2 and Table S1 (emm28, 89, 1, 4, 12, 3, 6, and 77), The sentence has been changed as follows: 

“The most “Prevalent” emm types were emm28, 89, 1, 4, 12, 3, 6, and 77, in agreement with those reported from other studies performed in developed countries.”

line 442

transposons => please correct

The correction has been made.

line 455

…a specific… factors… => please rephrase

The sentence has been rephrased as follows: 

“Herein, we failed to identify any specific risk factor (clinical or behavioural) that could explain the emergence of the genotype emm75 in a susceptible population.”

line 516

Ygout (?)

The capital letters for the name of the microbiologist have been changed.

figure 2

It would be nice if the percent sign could also be included at 0-53 and 54-76 (top left).

As suggested the percent sign has been included to 0-53 and 54-76 of the figure 2.

figure 3

It would be nice if in the middle column with the sporadic infections in figure 3a some more emm types could be marked in the figure.

As suggested in figure 3a we have indicated all sporadic emm types that have been isolated at least 3 times.

S3 figure

-To me the arrangement of the brackets for the representation of the age intervals seems to take some getting used to (see also lines 192-195).

All the open brackets have been modified (S3 figure and line 192-195) as indicated above.

-The percentages of 65% and 63% given in lines 194-195 do not correspond to those given in S3 Figure.

To more clarity we have corrected the percentages and rephrased the sentence by the following : 

“ Focusing on the overall 889 infection cases, the sex ratio (M/F) was 1.16 (54 % males); however, the rate of females was over-represented in the [30-40[ age-group (64% of females) while the percentage of males was higher in the [40-50[ age-group (68% of males) (S3 Fig).”

-For the age group of [30-40[, there should be a ‘-‘ instead of a ‘;’.

The correction has been made.

---

## [Editor Report · Decision Letter 1]

30 Nov 2020

PONE-D-20-31664R1

A prospective survey of Streptococcus pyogenes infections in French Brittany from 2009 to 2017: Comprehensive dynamic of new emergent emm genotypes.

PLOS ONE

Dear Dr. Kayal,

Thank you for submitting your manuscript to PLOS ONE. After careful consideration, we feel that it has merit but does not fully meet PLOS ONE’s publication criteria as it currently stands. Therefore, we invite you to submit a revised version of the manuscript that addresses the points raised during the review process.

I have some minor issues for the authors’ consideration:

References in abstract (line 37). Please delete.

Lines 73-75, “In contrast, in Africa and the Pacific region, the distribution of emm types exhibits a higher diversity explained by the non-observed dominant emm types [13].” Use of English.

Lines 79-80, “In tropical countries, the most circulating emm types of GAS are of emm pattern D (skin tropism) or E (both skin and pharyngeal tropism) (…)”. Use of English.

Lines 169-170, “Once validated, the dataset basis was completely anonymized.” The authors probably mean that the database was completely anonymized.

Lines 177-179, “Out of the 21 areas of French Brittany, and as expected, GAS isolates were predominantly recovered from patients living in Rennes and surrounding areas and then decreased gradually (S1 Fig)”. Ambiguous meaning.

Lines 180-182, “During the surveillance period, and as previously described, we also observed seasonal variation of the rate of infections (invasive and non-invasive), culminating in autumn/winter (S2 Fig)[23].” Perhaps “peaking”?

Lines 231-232, “(…) the final diagnosis was septic arthritis (n=14), isolated bacteraemia (12), central nervous system infections (2), primary peritonitis (2), and pericarditis (1).” Here and elsewhere if number of cases is meant then these should be indicated by n=.

Lines 358-361, “Finally, emm types clustered as a single protein, and for which it has been proposed that their M protein could have different immunological, structural, and functional characteristics were grouped in the same raw and encompassed “Prevalent” (emm6) and “Sporadic” (emm5, emm29, and emm105) emm types (Fig 3).” Replace raw with row?

Lines 368-369, “Deciphering the temporal dynamic of emm genotypes, we observed five “Emergent” emm types that belonged to emm-clusters that “Sporadic” but not “Prevalent” genotypes were members of.” Use of English.

Lines 436-437, “(…)(skin or generalist rather throat specialist  strains)(…)” missing “than”.

Lines 456-457, “This genotype upsurges last decades and has been linked recently to the emergence(…)”. Use of English.

I invite you to submit a revised version of the manuscript that addresses these points.

We look forward to receiving your revised manuscript.

Kind regards,

Jose Melo-Cristino, M.D., Ph.D.

Academic Editor

PLOS ONE

---

## [Author Response · Author response to Decision Letter 1]

1 Dec 2020

Response and modifications to each point raised by the academic editor and reviewer(s)

Issues for the authors’ consideration:

References in abstract (line 37). Please delete.

The references have been deleted. Then all the references have been reordered accordingly. 

Lines 73-75, “In contrast, in Africa and the Pacific region, the distribution of emm types exhibits a higher diversity explained by the non-observed dominant emm types [13].” Use of English.

The sentence has been changed by the following:

“In contrast, in Africa and the Pacific Islands, the distribution of emm-types is more diverse and does not show dominant emm-types.”

Lines 79-80, “In tropical countries, the most circulating emm types of GAS are of emm pattern D (skin tropism) or E (both skin and pharyngeal tropism) (…)”. Use of English.

The sentence has been changed by the following:

“In tropical countries, the most frequently isolated emm types of GAS belong to emm pattern D (skin tropism) or E (no specific tropism), as opposed to temperate regions where there are more strains of emm pattern A–C (pharyngeal tropism).”

Lines 169-170, “Once validated, the dataset basis was completely anonymized.” The authors probably mean that the database was completely anonymized.

“dataset basis” has been changed to “database”

Lines 177-179, “Out of the 21 areas of French Brittany, and as expected, GAS isolates were predominantly recovered from patients living in Rennes and surrounding areas and then decreased gradually (S1 Fig)”. Ambiguous meaning.

The sentence has been changed by the following:

“ The diagnosis of GAS infections was mainly performed within the UHC of Rennes, and explain that, over the 21 regions of French Brittany, the majority of GAS isolates was recovered from patients residing in Rennes and the neighbouring areas (Fig S1).

Lines 180-182, “During the surveillance period, and as previously described, we also observed seasonal variation of the rate of infections (invasive and non-invasive), culminating in autumn/winter (S2 Fig)[23].” Perhaps “peaking”?

“culminating” has been changed to “peaking”

Lines 231-232, “(…) the final diagnosis was septic arthritis (n=14), isolated bacteraemia (12), central nervous system infections (2), primary peritonitis (2), and pericarditis (1).” Here and elsewhere if number of cases is meant then these should be indicated by n=.

For all parentheses indicating the number of cases, an "n =" has been added

Lines 358-361, “Finally, emm types clustered as a single protein, and for which it has been proposed that their M protein could have different immunological, structural, and functional characteristics were grouped in the same raw and encompassed “Prevalent” (emm6) and “Sporadic” (emm5, emm29, and emm105) emm types (Fig 3).” Replace raw with row?

“raw” has been replaced with “row”

Lines 368-369, “Deciphering the temporal dynamic of emm genotypes, we observed five “Emergent” emm types that belonged to emm-clusters that “Sporadic” but not “Prevalent” genotypes were members of.” Use of English.

The sentence has been changed by the following:

“Deciphering the temporal dynamics of the emm genotypes in our studied population, we observed that the five "Emergent" emm types never belonged to clusters within which "Prevalent" genotypes have been identified.”

Lines 436-437, “(…)(skin or generalist rather throat specialist strains)(…)” missing “than”.

“than” has been added

Lines 456-457, “This genotype upsurges last decades and has been linked recently to the emergence(…)”. Use of English.

The sentence has been changed as follows:

” This genotype upsurges last decades and has recently been associated with the emergence of a new successful clade variant that has undergone several genetic modifications affecting known virulence factors.”

---

## [Editor Report · Decision Letter 2]

3 Dec 2020

A prospective survey of Streptococcus pyogenes infections in French Brittany from 2009 to 2017: Comprehensive dynamic of new emergent emm genotypes.

PONE-D-20-31664R2

Dear Dr. Kayal,

We’re pleased to inform you that your manuscript has been judged scientifically suitable for publication and will be formally accepted for publication once it meets all outstanding technical requirements.

Kind regards,

Jose Melo-Cristino, M.D., Ph.D.

Academic Editor

PLOS ONE
---

## [Editor Report · Acceptance letter]

7 Dec 2020

PONE-D-20-31664R2 

A prospective survey of *Streptococcus pyogenes* infections in French Brittany from 2009 to 2017: Comprehensive dynamic of new emergent emm genotypes. 

Dear Dr. Kayal:

I'm pleased to inform you that your manuscript has been deemed suitable for publication in PLOS ONE. Congratulations! Your manuscript is now with our production department. 

Kind regards, 

on behalf of

Prof. Jose Melo-Cristino 

Academic Editor

PLOS ONE